# Spinning Straw into Gold: Relabeling LLM Agent Trajectories in Hindsight for Successful Demonstrations

**Zichao Li**[1*]**, Gang Wu**[2]**, Zichao Wang**[2]**, Ruiyi Zhang**[2]**,**
**Wanrong Zhu**[2]**, Ryan A. Rossi**[2]**, Vlad I. Morariu**[2]**, Jihyung Kil**[2]

[1]Mila, McGill University      [2]Adobe Research
`zichao.li@mail.mcgill.ca, jkil@adobe.com`

## ABSTRACT

Large language model agents operate in partially observable, long-horizon settings where obtaining supervision remains a major bottleneck. We address this by utilizing a source of supervision overlooked in existing post-training methods: unintended yet successful goals embedded within agent rollouts. Specifically, we introduce Hindsight Supervised Learning (HSL), where an auxiliary LLM reviews each completed trajectory and relabels it with all of the natural-language goals the agent actually achieved. HSL then pairs the trajectory with its relabeled goals and uses these pairs for additional fine-tuning. To mitigate suboptimality in the relabeled data, we propose two learning techniques for HSL, irrelevant-action masking and sample reweighting. Our experiments show that HSL is flexible and compatible with existing post-training pipelines. It improves both SFT and DPO, with larger gains on long-horizon tasks with more diverse goal spaces. Moreover, HSL is sample-efficient: on ALFWorld, it surpasses baselines trained on the full dataset while using only one quarter of the ground-truth demonstrations.

## 1 INTRODUCTION

Large language model (LLM) agents extend foundation models (Bommasani et al., 2021) by enabling them to operate within interactive environments, including but not limited to autonomous web browsing and form completion (Zheng et al., 2024), tool-augmented question answering (Zhuang et al., 2023), software engineering (Jimenez et al., 2023), strategic decision-making in simulators and games (Fan et al., 2022), and high-level control of embodied robots (Li et al., 2024). Such agents are increasingly important because they bridge the gap between raw language models and practical, interactive intelligence. However, building effective LLM agents remains a challenging task. In most tasks, the dynamics of the underlying high-dimensional environment are complex and hidden, and the observations available to the agent reveal only partial information about it. Even so, the agent should plan over long horizons and choose actions whose effects are uncertain.

A variety of methods have been explored to train LLM agents, including supervised fine-tuning (SFT), direct preference optimization (DPO) (Rafailov et al., 2023), and other reinforcement learning (RL) techniques (Schulman et al., 2017; Liu et al., 2025). Among them, SFT is the most widely used, but it relies heavily on expert demonstrations, which are costly to collect and often lack diversity. More importantly, it does not fully leverage the trajectories generated by the agent itself. In contrast, RL can, in principle, exploit such data through trial and error. However, it requires discovering high-return trajectories before learning can proceed, which is challenging in long-horizon tasks with sparse rewards. The difficulty is further intensified when the agent's interaction with the environment is restricted, for example, by safety or privacy concerns in embodied or web agents (Tur et al., 2025).

The intuition behind this work, inspired by hindsight experience replay (Andrychowicz et al., 2017) in goal-conditioned RL, is that an LLM agent may accomplish unintended but valid tasks regardless of whether it completes the instructed one. Take Figure 1 as a conceptual example. When instructed to reach the blue flag, the agent might miss the target entirely and instead arrive at the red flag. Alternatively, it might reach the yellow flag before eventually reaching the

---

*Work done during an internship at Adobe Research.

blue one. Therefore, it is feasible to relabel the agent's trajectories with the unintended goals, such as the red or yellow flags, and then extract such unintended achievements from the agent's experience, turning them into successful demonstrations that can be consolidated for learning and performance gains. We assume that the relabeling task is easier than solving the original task, as it can be performed after the full trajectory is collected and does not require predicting environment dynamics, which are often among the hardest aspects of LLM agent tasks.

Instead, the problem reduces to inferring which tasks were actually accomplished by reasoning over the observations in hindsight, a perception and reasoning challenge that LLMs are well-suited to handle.

In view of this, we propose **H**indsight **S**upervised **L**earning (HSL) (Figure 2). HSL uses an auxiliary LLM to relabel agent-generated trajectories with the goals actually achieved by the agent and fine-tunes on these relabeled examples with SFT. Since the original trajectories are generated under different instructions, they may be suboptimal for the relabeled goals. We thus introduce two additional learning strategies, irrelevant-action masking and relabeled demonstration reweighting. These strategies help agents learn more effectively to follow natural language instructions and reproduce successful behaviors.

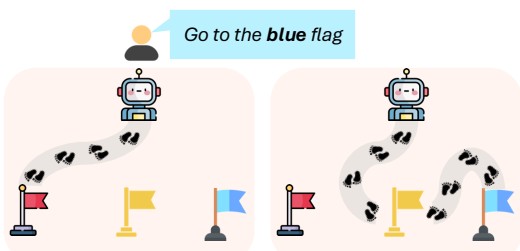

Figure 1: Toy example. Given the instruction "Go to the blue flag", the agent may instead reach the red flag or visit the yellow flag along the way. While these trajectories are incorrect or not optimal for the original instruction, they can be relabeled as successful demonstrations for the goals the agent actually achieved.

In sum, our contributions are threefold:

- We propose a novel post-training method for LLM agents, Hindsight Supervised Learning (HSL), which iteratively mines successful demonstrations by relabeling agent trajectories and fine-tunes the agent on the relabeled data.

- Based on operational insights from our theoretical analysis, we introduce two simple yet effective learning methods, irrelevant-action masking and sample reweighting, which are further validated through ablation studies.

- Our experiments on ALFWorld (Shridhar et al., 2020), PlanCraft (Dagan et al., 2024) and WebShop (Yao et al., 2022) demonstrate that (1) HSL is **compatible** with existing post-training methods such as SFT and DPO, yielding notable improvements (e.g., by $8\% - 32\%$ success rate in ALFWorld), (2) HSL is sample-efficient. For instance, with less than a quarter of ground-truth demonstrations, it outperforms baseline methods trained on the full dataset.

## 2 RELATED WORK

### 2.1 LLM AGENTS

Early language agents were developed for artificial text-based interactive environments (Chevalier-Boisvert et al., 2018), where agents follow natural language instructions to change states and achieve goals. This shift in focus transformed NLP from static prediction to sequential decision-making. With the rapid advancement of language models, LLM agents now operate in more realistic and complex domains. These include embodied agents (Li et al., 2024); GUI agents (Yao et al., 2022; Zhou et al., 2024; Xie et al., 2024); and code agents (Jimenez et al., 2023).

Post-training methods for these agents follow two main approaches. Supervised fine-tuning (SFT) on demonstrations is effective but costly and limited by coverage. Preference- and RL-based training, such as PPO (Schulman et al., 2017; Hu et al., 2025), DPO (Rafailov et al., 2023; Song et al., 2024), aims to improve agents with weaker supervision, yet often struggles under sparse, delayed feedback and long horizons. To mitigate sparsity, recent approaches synthesize feedback using heuristics or

learned judges (Da et al., 2025) and conduct intensive interactions with the environment to obtain denser intermediate rewards (Xiong et al., 2024).

By contrast, our work proposes an alternative approach: maximizing the number of successful demonstrations through agent experiences. Nevertheless, this approach is complementary to existing learning algorithms, and we demonstrate that it can provide an additive improvement when combined with different post-training methods.

Another line of research synthesizes large-scale training data with heuristics (Xu et al., 2024), some of which also involve generating a label conditioned on agent trajectories (Murty et al., 2024; Sun et al., 2024). While these methods assume a single goal or instruction per trajectory, we propose to mine all achieved goals within the trajectory. More importantly, these data synthesis methods generate training data using a different model from the target LLM agent, which is fine-tuned with the synthesized dataset fixed. By contrast, we continually refresh the relabeled buffer with the same target agent. This on-policy, continuously updated supervision aligns the hindsight distribution with the agent's evolving occupancy, improves coverage where the expert policy has support, and mitigates the drift induced by fixed synthetic datasets. As shown in Section 3.4 and Section 4, this design both tightens the expert–agent gap in analysis and translates to consistent empirical gains over fixed-data synthesis baselines.

## 2.2 Goal Relabeling for Goal-conditioned RL

The idea of goal relabeling was first proposed in hindsight experience replay (HER) for robotic tasks (Andrychowicz et al., 2017). HER relabels episodes with goals that are actually achieved, enabling efficient learning from sparse, binary rewards. Since then, HER has become a workhorse for multi-goal RL and robotics. Several extensions refine how experiences are selected and weighted, such as countering bias in relabeled experiences by applying more aggressive hindsight rewards (Lanka & Wu, 2018), scheduling replay toward experiences closer to actual goals while maintaining diversity (Fang et al., 2019), relabeling trajectories explicitly assemble success from multiple failures (Fang et al., 2018). Later, Cideron et al. (2020) extends this idea to language-conditioned agents, learning an instruction generator to map the reached states to text-based instructions. While this approach, based on HER, trains the agent on the relabeled experiences with the original offline RL objective, GCSL (Ghosh et al., 2019) instead applies supervised learning to relabeled successes. Nonetheless, a complementary robotics line learns from play data by retrospectively specifying goals within unlabeled teleoperation and training goal-conditioned policies (Lynch et al., 2020; Cui et al., 2022).

Our work differs from prior goal-conditioned RL research using goal relabeling techniques in several ways. First, most existing methods assume the agent has direct access to the whole state of the environment. In contrast, we address a more complex setting where the agent only receives partial observations, so goal relabeling must be inferred from multiple key observations. Second, methods such as Cideron et al. (2020); Ghosh et al. (2019) typically relabel only the final goal of failed trajectories. In contrast, we relabel all goals achieved along the trajectory, even when the instructed goal is eventually reached. Third, while HER and its variants train agents using offline RL, we apply SFT and propose two additional techniques to drive the agent towards optimal policy. Finally, our approach focuses on enhancing LLM agents by leveraging the reasoning abilities of LLM for the relabeling process.

More recently, Zhang et al. (2023) proposed rewriting queries to better align with LLM-generated answers for BigBench reasoning tasks (Srivastava et al., 2023), such as logical deduction and word sorting. Although the high-level idea is related, our work focuses on agentic POMDP tasks and introduces distinct techniques.

## 3 Methods

In this section, we present Hindsight Supervised Learning (HSL), a new post-training method for LLM agents. We begin with the background and problem formulation, then outline the three main stages of HSL: trajectory collection, relabeling trajectories with an auxiliary LLM to produce successful demonstrations, and learning from these relabeled demonstrations. We then explain the relabeling process in detail, the core component of HSL, which consists of two steps: goal identification and

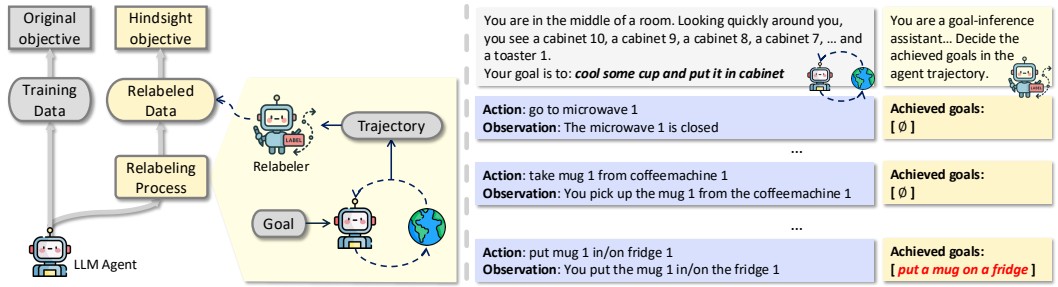

Figure 2: Left: Overview of the existing training pipeline with HSL. Right: Example of the relabeling process. The relabeler assigns new goals to the agent's trajectory based on the agent's achievements.

action relevance labeling. Finally, we present two additional learning techniques that further optimize the agent, relevance-based loss masking and relabeled demonstration reweighting.

## 3.1 BACKGROUND

We model an LLM agent as operating in a partially observable Markov decision process (POMDP). At each step $t$, the environment produces an observation $o_t$ that reveals only part of the underlying state $s_t$. The transition from the high-dimensional $s_t$ to $s_{t+1}$ induced by action $a_t$ is assumed to be unknown. Conditioned on the natural language instruction $I$, the trajectory history $\tau_{t-1} = \{(o_i, a_i)\}_{i=1}^{t-1}$, and the new observation $o_t$, the agent selects an action $a_t$ that maximizes its learned conditional distribution $\pi_\theta(a_t|\tau_{t-1}, o_t, I)$. An reward function $R_T(s_T, g(I))$ measures how well the final state achieves the goal specified by $I$. We also assume access to a general textual description of the goal space $\mathcal{G}$. The task is to find the best agent that maximize the expected reward: $\max_\theta \mathbb{E}_{\tau \sim \pi_\theta}[R_T]$. Typically, the agent is trained using offline ground-truth demonstrations. A simple and straightforward way of learning from demonstrations is supervised fine-tuning (SFT). In addition, one may use RL methods such as PPO (Schulman et al., 2017) and GRPO (Shao et al., 2024).

## 3.2 FRAMEWORK OF HINDSIGHT SUPERVISED LEARNING

In general, our proposed HSL framework adds a parallel branch with three key steps: trajectory collection, relabeling, and learning. This branch can be combined with arbitrary existing training pipelines, including SFT and DPO. We describe each step below.

**Trajectory collection.** At each training step, we sample $b$ goals expressed in natural-language instructions $I \sim D_{\text{train}}$ from the training set. The agent then rolls out by following $I$ through interaction with the environment. Specifically, the agent selects an action $a_t \sim \pi_\theta(a|\tau_{t-1}, o_t, I)$ and receives the next observation $o_{t+1}$. This process continues until the task terminates or the maximum step limit $T$ is reached. We then collect trajectories $(\tau_T, o^*)$, including the final observation $o^*$, for relabeling.

**Relabeling process.** We use an auxiliary LLM $M$ to revisit each collected trajectory and identify the actual $K$ goals $(K \geq 0)$, represented as instructions $\{I'_1, \ldots, I'_k\}$, that the agent successfully achieved. From these, we extract pairs of relabeled instructions and their corresponding trajectories $(I', \tau')_k$ as successful demonstrations. The relabeling process is described in detail in Section 3.3. The resulting demonstrations are stored in a dynamic buffer $D'$ of size $N$.

**Learning from relabeled demonstrations.** We improve the LLM agents with the relabeled demonstrations. At each optimization step, we randomly sample a batch of relabeled demonstrations from $D'$ and calculate $\mathcal{L}_\theta^{\text{HSL}}$. To further improve agent optimality, we propose two techniques: irrelevant action masking and demonstration reweighting. We detail both of them and the concrete loss function $\mathcal{L}_\theta^{\text{HSL}}$ on Section 3.4.

This HSL training pipeline on the relabeled demonstrations $D'$ can be combined with a wide range of existing post-training methods, including SFT and DPO, on $D_{\text{train}}$. Notably, the relabeling process

avoids reliance on any ground-truth actions or reward signal, allowing the relabeling model to operate in an almost unsupervised manner.

### 3.3 RELABELING FOR SUCCESSFUL DEMONSTRATIONS IN HINDSIGHT

Next, we provide a detailed explanation of the relabeling process, which is the core component of HSL. It consists of two steps: goal identification and action relevance labeling.

**Goal identification.** While it is hard to foresee which goals will be achieved after an action $a_t$, reasoning from the resulting observation $o_{t+1}$ makes the task more manageable. Typically, $o_{t+1}$ reveals whether $a_t$ had any effect on the environment and, if so, what that effect was. One can also infer the long-term impact of $a_t$ by examining subsequent observations.

We therefore employ an external LLM $M$ to revisit each trajectory and infer a running list of achieved goals represented as instructions $\mathcal{I}' = I'_1, \ldots, I'_K$ along the trajectory. Although $M$ could be trained or adapted for this task, we instead rely on its zero-shot reasoning ability and prompt it to infer the achieved goals in one pass: $\{I'_1, \ldots, I'_K\} = M_{\text{inst}}(\tau_T, o^*)$, where `inst` denotes the system prompt (provided in Appendix C) describing the space of valid goals, and $o^*$ is the final observation.

Figure 2 (right) illustrates a concrete example: the agent has missed the original goal *cool some cup and put in cabinet*, but achieved an uninstructed one *put a mug in a fridge*. In some cases, the agent may have multiple achievements during the trial, which can be independent of one another. If the agent fails to achieve any meaningful goal in $\tau_T$, we discard the trajectory. We then extract each pair of a relabeled goal and its corresponding trajectory segment $(\tau_{S(I'_k)}, I'_k)$ as a successful demonstration, where $S(I'_k)$ is the step where $I'_k$ is newly accomplished.

**Action relevance labeling.** Since the trajectory $\tau_{S(I'_k)}$ is originally collected for instruction $I$, some actions in $\tau_{S(I'_k)}$ may be irrelevant to achieving the relabeled goal $I'k$. For each pair $(\tau_{S(I'_k)}, I'_k)$, we reuse $M$ to infer a label $z_{u \in 1, \ldots, S(I'_k)}$ for each action $a_u$ in $\tau_{S(I'_k)}$, indicating its relevance to $I'_k$: $z_{1:S(k)} = M_{\text{relevance}}(\tau_{S(k)}, o_{S(k)+1}, I'_k)$, where `relevance` is the system prompt shown in Appendix C. The inferred $z_{1:S(k)}$ is subsequently used to train the LLM agents with the relabeled demonstrations.

While existing hindsight generation in RL mostly relabels a single goal on the failed trajectory (Cideron et al., 2020; Ghosh et al., 2019), our relabeling model $M$ identifies all valid instructions achieved in $\tau$. This design is motivated by two reasons. First, even in successful trajectories, the agent may accomplish additional tasks at intermediate steps. We find that the LLM agent benefits from learning these uninstructed tasks. Second, HSL is less dependent on explicit reward signals, which are often noisy or difficult to obtain in LLM agent tasks such as web-based agents.

At the end of relabeling, we extract each triplet of $(\tau_{S(k)}, I'_k, z_{1:S(k)})$ and append it to the relabeled demonstration buffer $D'$.

### 3.4 THEORETICAL ANALYSIS AND LEARNING TECHNIQUES

We would like to analyze how learning from the relabeled demonstrations bridges the gap between the LLM agent and the optimal (expert) policy $\pi^*$. We define the *hindsight expert* induced by agent trajectories and the relabeling model $M$ as:

$$\pi_H(a \mid \tau, o, I') = \Pr_{\pi_\theta}\left[a_{|\tau|+1} = a \,\big|\, \tau_{|\tau|} = \tau, \; o_{|\tau|+1} = o, \; S_M(I') = |\tau| + 1\right] \quad (1)$$

recall that $S_M(I')$ denotes that the first timestep $I'$ is achieved. Accordingly, we define $\kappa_{E \leftarrow H}$ as the occupancy coverage ratio of $\pi^*$ and $\pi_H$, and $\delta_E$ as the discrepancy between $\pi^*$ and $\pi_H$:

$$\kappa_{E \leftarrow H} \triangleq \max_{(\tau,o,I)} \frac{\rho_{\pi^*}(\tau,o,I)}{\rho_{\pi_H}(\tau,o,I)} \in [1, \infty); \quad \delta_E = \mathbb{E}_{(\tau,o,I) \sim \rho_{\pi^*}}\left[\left\|\pi_H(\cdot \mid \tau,o,I) - \pi^\star(\cdot \mid \tau,o,I)\right\|\right],$$

where $\rho_\pi$ is the occupancy measure of $\pi$ over $(\tau, o, I)$.

Our target is to minimize the following discrepancy between agent $\pi_\theta$ and optimal policy $\pi^*$:

$$\Delta(\theta) \triangleq \mathbb{E}_{(\tau,o,I) \sim \rho_{\pi_\theta}}\left[\left\|\pi_\theta(\cdot \mid \tau,o,I) - \pi^\star(\cdot \mid \tau,o,I)\right\|\right].$$

**Theorem 1** (Upper Bound of Expert-Agent Discrepancy). *Under the setup above,*

$$\Delta(\theta) \leq \mathbb{E}_{(\tau,o,I)\sim\rho_{\pi^*}}\left[ D_{\mathrm{KL}}(\pi^\star(\cdot \mid \tau, o, I) \,\|\, \pi_\theta(\cdot \mid \tau, o, I)) \right]$$
$$+ C_T\, \kappa_{E\leftarrow H}\, \sqrt{\tfrac{1}{2}\, \mathbb{E}_{(\tau,o,I')\sim\rho_{\pi_H}}\left[ D_{\mathrm{KL}}(\pi_H(\cdot \mid \tau, o, I') \,\|\, \pi_\theta(\cdot \mid \tau, o, I')) \right]} \;+\; C_T\, \delta_E\,. \tag{2}$$

*where $C_T = 2(T-1)$.*

Appendix D provides the detailed proof. The key takeaways of Theorem 1 are as follows. The upper bound on the discrepancy between $\pi_\theta$ and $\pi^*$ decreases when we apply SFT to both ground-truth demonstrations (first term) and relabeled demonstrations (second term). The gap shrinks further as the occupancy coverage and the optimality of $\pi_H$ improve. In practice, this can be achieved by using a stronger relabeling model and, more importantly, by continually updating the relabeled buffer $D'$ so that $\pi_H$ is constructed on-policy. As the agent succeeds more often, $\rho_{\pi_H}$ concentrates its mass where $\rho_{\pi^*}$ has mass. We also prove a corollary in Appendix E showing that SFT on relabeled demonstrations strictly tightens the expert–agent discrepancy (Equation (2)). Guided by these insights, we train on relabeled demonstrations with SFT and introduce two simple, effective learning techniques.

**Demonstration reweighting.** Some instructions $I'$ are inherently easier and thus appear disproportionately in $D'$, which may bias the agent toward solving only trivial tasks, leading to a bad coverage $\kappa$. To mitigate this issue, we sample demonstrations $d \in D'$ with a weight $w_d$ that prioritizes learning from trajectories that solve more difficult tasks more optimally. Concretely, $w_d$ is calculated by: $w_d = (\frac{n_d}{T_d})^\alpha \cdot n_d$, where $n_d$ is the number of actions associated with the relevance label $z_u = 1$, and $T_d$ is the total number of actions in $d$. The ratio $\frac{n_d}{T_d}$ reflects how optimally the task is solved, while $n_d$ serves as a proxy for task difficulty. The hyperparameter $\alpha$ balances these two factors.

**Irrelevant action masking.** Because the original trajectories are collected under instructions different from the relabeled $I'$, naively applying SFT on all actions in each trajectory may imitate a sub-optimal hindsight expert with large $\delta_E$. Therefore, we mask out the loss for actions labeled irrelevant ($z_t = 0$). And hence, the training objective on relabeled demonstrations is defined as:

$$\mathcal{L}_\theta^{D'} = \mathbb{E}_{d'=(\tau,I',z)\sim P(\cdot)}\left[ \frac{1}{T}\sum_{t=1}^{T} -z_t \cdot \log P_\theta(a_t|\tau_{t-1}, o_t, I') \right], \tag{3}$$

where $P(d') = \frac{w'_d}{\sum_{d\in D'} w_d}$. Suggested by Theorem 1, we further incorporate the SFT loss on ground-truth demonstrations $D_{\mathrm{train}}$ using a mixture weight $\lambda$, and thus the resulting learning objective is:

$$\mathcal{L}_\theta^{\mathrm{HSL}} = \lambda\mathcal{L}_\theta^{D'} + (1-\lambda)\mathbb{E}_{d=(\tau,I)\sim D_{\mathrm{train}}}\left[ \frac{1}{T}\sum_{t=1}^{T} -\log P_\theta(a_t|\tau_{t-1}, o_t, I) \right]. \tag{4}$$

Similar to many existing methods of post-training LLM agents (Song et al., 2024), HSL requires ground-truth demonstrations for stable learning. However, as we will show later in the experiment, HSL is sample efficient and achieves performance that matches or exceeds baseline methods while using much less $D_{\mathrm{train}}$.

## 4 EXPERIMENT

### 4.1 SETUP

**Tasks and Evaluation Metrics.** We evaluate on three well-adopted agentic benchmarks with different levels of task diversity. **ALFWorld** (Shridhar et al., 2020) is an embodied agent benchmark in which the LLM agent navigates rooms and completes household tasks to satisfy a natural language goal (e.g., "put some vase in safe"). The evaluation set consists of a *seen* split (new tasks within scenes present in the training set) and an *unseen* split (rooms or layouts absent from the training set). A task is considered successful only if all goal conditions are met. Following Song et al. (2024), we report the average *Success Rate*. **PlanCraft** (Dagan et al., 2024) is a Minecraft crafting benchmark that tests an agent's planning in a simplified crafting UI with text-only observations. Given a natural

language goal item, an initial inventory and the recipe of the goal item, the agent must craft the target by moving items between inventory slots and the crafting grid and by smelting items as needed. We exclude the impossible subset and report the average *Success Rate*. **WebShop** (Yao et al., 2022) is a web agent benchmark in which an agent follows a natural language instruction to navigate a simulated e-commerce site and purchase a product. Each episode ends upon purchase and returns a dense reward $r \in [0, 1]$ based on the type matching, the coverage of the requested attributes/options, and the price constraint. We report *Task Score* ($100\times$ average reward). ALFWorld features longer horizons ($T = 40$), a diverse set of valid goals, and multiple achievable goals per episode. In contrast, PlanCraft and WebShop each consist of a single task type with shorter horizons ($T = 30$ and $T = 10$, respectively). Moreover, each trajectory in WebShop can achieve only one goal. Therefore, we expect HSL to yield larger gains on ALFWorld, while PlanCraft and WebShop primarily test its robustness in a narrower task space.

**Implementation.** We employ Llama-3.2-1B (Dubey et al., 2024) as an agent model and Llama-3.3-70B (Dubey et al., 2024) as a relabeling model. We set $\lambda = 0.3$ and $\alpha = 0.8$, and maintain the relabeled dataset $D'$ as a queue of size 100. After each optimization step, HSL collects and relabels $b = 18$ trajectories, which are appended to $D'$ while the oldest entries are removed once the limit is reached. Additional implementation details are provided in Appendix F.

**Baselines.** We evaluate HSL as an add-on to SFT and Exploration-based Trajectory Optimization (ETO) (Song et al., 2024). SFT fine-tunes the agent model on ground-truth demonstrations from the original datasets, while ETO applies DPO (Rafailov et al., 2023) to agent tasks, using ground-truth demos as preferred samples and agent-generated failures as dispreferred samples. We also include SELFIMIT (Shi et al., 2023), which fine-tunes the agent on its own successful trajectories. To demonstrate that the gains stem from our relabeling approach rather than merely using a powerful external LLM, we include baseline methods that also use Llama-3.3-70B. Concretely, REACT (Yao et al., 2023) directly applies Llama-3.3-70B for reasoning and action selection. BEHAVIORCLONE uses Llama-3.3-70B to synthesize demonstrations and then fine-tunes the agent on the union of ground-truth and synthetic data, following Zeng et al. (2024). BAGEL (Murty et al., 2024) is an offline data synthesis method that generates trajectories with the agent and uses Llama-3.3-70B to label and filter them; subsequently, the agent is fine-tuned on the synthesized and ground-truth data.

## 4.2 MAIN RESULTS

Table 1 presents the performance of our proposed HSL and prior related methods across all three benchmarks. HSL improves both SFT and DPO significantly and consistently with much larger gains on ALFWorld, which involves longer tasks and a larger set of valid goals. In addition, the improvement on WebShop is smaller than that on PlanCraft, likely because WebShop allows only one achievable goal per trajectory, whereas LLM agents can craft multiple items within a single task in PlanCraft. Although REACT and BEHAVIORCLONE use the external LLM (Llama-3.3-70B) to enhance the agent, they perform substantially worse than SFT+HSL, validating our claim that predicting actions for an agentic task itself is more challenging than relabeling. BAGEL also improves SFT on ALFWorld but still underperforms compared to SFT+HSL, showing the importance of continuously updating the relabeled demonstrations for the target agent during training. This finding echoes our analysis in Section 3.4, which suggests that the relabeling process can benefit from the evolved agent. Finally, SELFIMIT fails to improve upon SFT, underscoring the importance of mining all successful demonstrations, even for *uninstructed* goals. To further verify the robustness of our results, we rerun all fine-tuning experiments on ALFWorld and WebShop using different random seeds and report the results in Appendix G.

**Sample efficiency.** To assess the sample efficiency of HSL, we evaluate how the number of ground-truth demonstrations affects the performance of HSL and the post-training baselines on ALFWorld and WebShop. As shown in Figure 3, adding HSL to either SFT and DPO consistently and substantially increases success rates on ALFWorld, given the same ground-truth data budget as the SFT and DPO baselines. Notably, the improvement is particularly larger in the *Unseen* split, where HSL nearly doubles SFT at 800 demonstrations and doubles DPO at 1,600 demonstrations. This again highlights that HSL facilitates generalization. More importantly, HSL, which uses less than one quarter of the ground-truth data, surpasses SFT-only or DPO-only models trained on the full dataset.

| Method | ALFWorld | | PlanCraft | WebShop |
|---|---|---|---|---|
| | seen | unseen | | |
| REACT (Yao et al., 2023) | 33.57 | 20.90 | 59.17 | 48.37 |
| BEHAVIORCLONE (Zeng et al., 2024) | 83.57 | 88.81 | 64.38 | 65.19 |
| BAGEL (Murty et al., 2024) | 84.29 | 91.79 | 69.17 | 62.18 |
| SELFIMIT (Shi et al., 2023) | 84.29 | 76.87 | 56.25 | 58.37 |
| SFT | 82.14 | 78.36 | 70.00 | 63.81 |
| DPO (Song et al., 2024) | 85.71 | 82.84 | 71.25 | 69.54 |
| SFT+HSL (Ours) | **93.57** | **97.76** | 75.12 | 66.97 |
| DPO+HSL (Ours) | 92.86 | 94.78 | **75.42** | **70.52** |

Table 1: Performance on ALFWorld, PlanCraft, and WebShop.

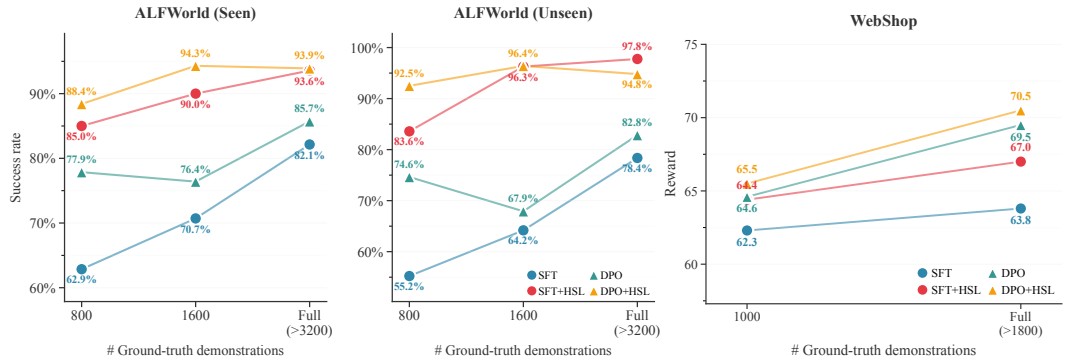

Figure 3: Sample efficiency of HSL with different post-training methods.

For example, DPO+HSL reaches $92.5\%$ with just 800 ground-truth demonstrations on ALFWorld (Unseen), whereas DPO attains only $82.8\%$ even with more than 3,200 ones. On WebShop, the improvements are smaller but follow trends similar to those in ALFWorld across data sizes and baselines. This suggests that HSL is particularly effective for open-ended tasks with diverse goal types, such as ALFWorld, where agents are more likely to "accidentally" accomplish uninstructed tasks and therefore benefit more from relabeling.

## 4.3 ABLATION STUDIES

We conduct ablation studies to quantify the contribution of each component and technique in HSL. In particular, we evaluate RELABELFAILURE, which only relabels the final achieved goal on failed trajectories and resembles existing hindsight generation methods in the RL literature (Cideron et al., 2020; Ghosh et al., 2019); UNIWEIGHT, which samples relabeled demonstrations uniformly without any reweighting; and NOMASK, which does not apply the action-irrelevant masking.

Figure 4 presents all variants, including the complete model SFT+HSL trained with different numbers of ground truth demonstrations on ALFWorld. Removing any component reduces performance, though the drop from removing demonstration reweighting (UNIWEIGHT) is smaller in the *Seen* split. In the *Unseen* split, UNIWEIGHT is markedly worse, which suggests that increasing expert–hindsight occupancy coverage in Theorem 1 facilitates generalization. With the fewest ground truth demonstrations, NOMASK shows the most significant decline, likely because a weaker base agent executes many back and forth actions within a trial, for example repeatedly visiting a receptacle, which are not helpful for the relabeled goal. This highlights the need for a stronger hindsight expert who proposes more optimal actions. RELABELFAILURE is consistently and substantially worse than the full model. In particular, it does not benefit from additional ground truth demonstrations, mainly because a more potent base agent trained with more demonstrations produces fewer failed trajectories. It verifies our decision to leverage all the intermediate goals that have been achieved.

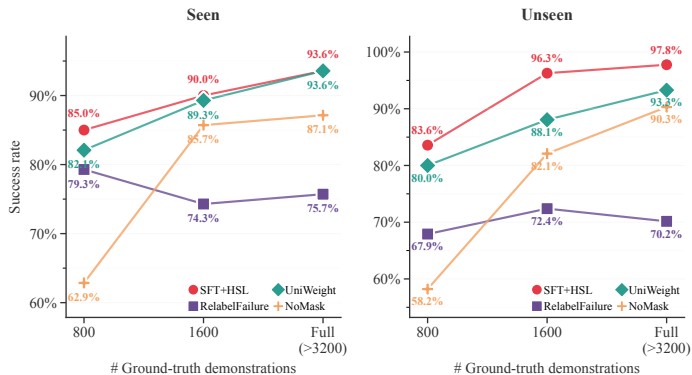

Figure 4: Ablation studies on ALFWorld.

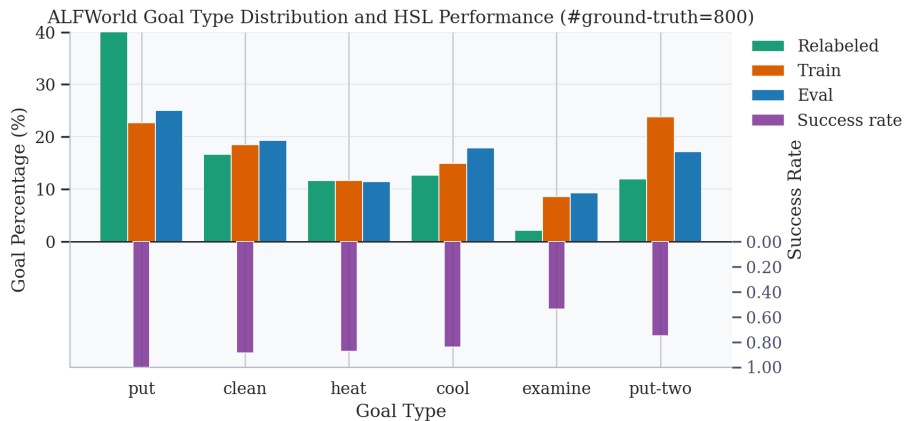

Figure 5: Top: Distribution of the goal types in relabeled and ground-truth demonstrations. Bottom: Success rate of SFT+HSL across different goal types.

## 4.4 ANALYSIS: UNDERSTANDING THE EFFECTIVENESS AND LIMITATIONS OF HSL

To gain a deeper understanding of the relabeled demonstrations and how learning from them helps, we conduct quantitative and qualitative analyses on ALFWorld, which includes diverse goal types. Because the model trained on the full dataset achieves near-perfect success rates across all tasks in ALFWorld, we focus on SFT+HSL trained with only 800 ground-truth demonstrations. First, we compare the distribution of goal types in the relabeled demonstrations with that of the ground-truth data. As shown in Figure 5, the most augmented goal in the relabeled set is put, and the agent correspondingly achieves perfect performance on put.

For clean, heat, and cool, the proportions in the relabeled data are close to those in the ground truth, and the agent achieves success rates above 80%. In contrast, examine and put-two are underrepresented in the relabeled demos. While put and put-two appear at similar rates in the ground truth, the relabeled demos contain roughly twice as many put as put-two. Moreover, examine accounts for less than 5% of the relabeled set. This pattern likely arises because put is a subtask of put-two, and examine is the most long-tailed goal in the training data (under 10%), such that the agent rarely

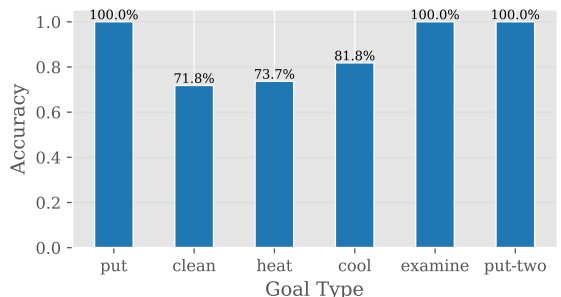

Figure 6: Accuracy of relabeling per goal type.

```
Example of Relabeled Demonstrations (Correct)

Relabeled Goal: put a clean cloth in toilet
...
Step 3
  Action: take cloth 2 from countertop 2
  Observation: You pick up the cloth 2 from the
  countertop 2.
  Relevant to Goal: True
...
Step 5
  Action: clean cloth 2 with sinkbasin 1
  Observation: You clean the cloth 2 using
  the sinkbasin 1.
  Relevant to Goal: True
...
Step 7(the last step)
  Action:  put cloth 2 in/on toilet 1
  Observation: You put the cloth 2 in/on the
  toilet 1.
  Relevant to Goal: True
```

```
Example of Relabeled Demonstrations (Incorrect)

Relabeled Goal: put a cool tomato in fridge
...
Step 30
  Action: take tomato 1 from countertop 1
  Observation: You pick up the tomato 1 from
  the countertop 1.
  Relevant to Goal: True

Step 31
  Action: go to fridge 1
  Observation: The fridge 1 is open. In it,
  you see a plate 2, and a potato 3.
  Relevant to Goal: True

Step 32(the last step)
  Action: cool tomato 1 with fridge 1
  Observation: You cool the tomato 1 using
  the fridge 1.
  Relevant to Goal: True
```

Figure 7: Two examples (correct and incorrect) of relabeled demonstrations in ALFWorld. Each shows agent trajectories with the corresponding outputs (relabeled goals and action relevance) by the relabeler. In the right example, the relabeler makes a mistake: the agent cools a tomato but *never* places it in the fridge, so only a subtask of the relabeled goal is satisfied.

achieves it by chance. As a result, the agent trained with the relabeled demos records its lowest success on `examine` at $54.85\%$, and its performance on `put-two` lags notably behind `put`.

We further assess the quality of the relabeled data. We randomly sample 200 relabeled demonstrations and manually verify their labels. Of these, 180 are correct, yielding an accuracy of $90\%$, which helps explain the strong performance of the HSL-trained LLM agent. Figure 6 shows the accuracy for each goal type. Notably, the relabeling is performed in a zero-shot setting, yet it is substantially more accurate than predicting actions with in-context examples (REACT). This observation further supports our hypothesis that relabeling trajectories in hindsight is easier than executing the task itself. Two illustrative cases are shown in Figure 7.

> **Takeaway**
>
> Taken together with our theoretical (Section 3.4) and empirical results, these findings indicate that HSL narrows the gap between the agent and the expert policy in a sample-efficient manner by transforming noisy agent trajectories into high-quality demonstrations from a hindsight expert. Nevertheless, the magnitude of the improvement depends on the hindsight expert's optimality and coverage, as well as task characteristics such as diversity and the number of tasks per trajectory.

## 5 CONCLUSION

We present Hindsight Supervised Learning (HSL), a sample-efficient learning framework that relabels agent trajectories with goals actually achieved and learns from these relabeled demonstrations. By incorporating irrelevant-action masking and reweighting, HSL enhances both the coverage and quality of relabeled data. Comprehensive experiments on three agentic benchmarks, ALFWorld, PlanCraft and WebShop, show that HSL consistently boosts SFT and DPO while reducing reliance on ground-truth demonstrations, with larger gains on long-horizon tasks with diverse goal spaces. These findings show that hindsight relabeling is an efficient way to leverage LLMs' reasoning capabilities while avoiding the need to model environment dynamics. As future work, we plan to extend this framework to more open-ended environments and multimodal agents. Nevertheless, our work still has limitations, which we discuss in Appendix B.

## ACKNOWLEDGMENT

We thank Tong Sun for discussions, as well as the anonymous reviewers and ACs for their feedback and comments.

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

## A  Language Model Usage Statement

During the preparation of the manuscript, we have used LLM to polish the draft and fix grammar issues. LLM did not participate in the ideation of the project, such as problem formulation and methodology.

## B  Limitations

**Limitations of the Proposed Method.**    Our method, in its current form, uses an external LLM for zero-shot relabeling without fine-tuning, so it depends on the reasoning ability of powerful models. This design lets us leverage state-of-the-art proprietary LLMs, and our experiments show that using them for relabeling yields markedly better final performance than asking the same models to solve the tasks directly. For more complex tasks, however, the relabeling model may require further adaptation and optimization. Jointly learning the relabeling model and the LLM agent is left for future work.

Second, we label each action in a trajectory as either relevant or not to the relabeled goal, which assumes a black-and-white notion of relevance. In complex settings an action can be relevant yet suboptimal for the goal. For example, purchasing multiple items one by one is less efficient than placing them all in a cart and paying once. A more fine-grained notion of relevance may therefore be needed, which would entail fine-tuning the relabeling model in a role akin to a reward or value function.

Third, as shown in the experiments, although our demonstration reweighting improves agent performance and many goal types are well represented in the relabeled demonstrations, the resulting distribution does not fully match the goal-type distribution in the original dataset. Our method chiefly exploits exploration in a sample-efficient way, so further work is needed to increase the diversity of exploration.

Fourth, while HSL is sample-efficient and reduces reliance on ground truth demonstrations, it unavoidably increases training compute because it uses an external relabeling model. The inference cost stays the same. This mirrors a broader trend in LLM agents: investing more compute during training (GPU hours and LLM tokens) to improve LLM agent performance (Lightman et al., 2023; Song et al., 2024). We believe that this is a valid contribution since collecting human demonstrations for agent tasks is costly and hard to scale.

Last, although the relabeling process and learning from relabeled demonstrations are unsupervised or self-supervised, HSL still requires ground truth demonstrations to stabilize training by aligning the agent's support with the expert's. This limitation is shared by many other post-training methods. Future work should develop stronger exploration strategies that enable HSL without any ground-truth demonstrations.

**Limitations of Experiments.**    We use Llama-3 as the base LLM agent and evaluate on three standard agentic benchmarks, ALFWorld, PlanCraft and WebShop. Although we conduct comprehensive ablations and analyses, we do not cover stronger base LLMs or additional agentic tasks. It is mainly due to the limit of time and computational resource as well as lack of task-specific ground-truth demonstrations. For many GUI-agent benchmarks, for instance, either an interactive environment or ground-truth demonstrations are missing. When they are available, the environments are often resource-intensive to set up.

Second, as discussed in the experiments, the gains from HSL are smaller for short-horizon tasks with a narrow goal space. We expect larger benefits in more open-ended settings with diverse goals, where LLM agents must adapt to broader task distributions.

Third, the tasks adopted in our experiments contains predefined goal spaces, enabling the relabeling model to determine whether there is any valid goal achieved and what it is. This setting is common in existing LLM agent benchmarks, but it may not hold in open-world environments. Future work can explore relabeling without a predefined goal space, such as deriving goal space from a set of user instructions or based on heuristics.

Last, most of existing LLM agent benchmarks, such as the ones adopted in our experiments, assumes all goals are equally valuable and safe. However, in complex real-world scenarios, the unintended

goals may have low value for the users or even be malicious. To address these issues, future work can score the utility of unintended goals by comparing it with the user instructions, and adopt safeguard models to filter out or penalize the malicious goals.

## C    PROMPTS FOR RELABELING PROCESS

Figure 8 and Figure 12 show the goal relabeling prompts for ALFWorld and WebShop, respectively. Figure 9 and Figure 13 are the action relevance labeling prompts for ALFWorld and WebShop. Because the crafted item in PlanCraft can be read directly from the returned text-based observation, we do not use the relabeling LLM to infer the crafted item. Instead, we use it to infer another part of the input: the recipe used to craft the goal item. Figure 10 presents the corresponding prompt. Figure 11 is the action relevance labeling prompt for PlanCraft.

## D    PROOF OF THEOREM 1

**Preliminaries and notation.**    For simplicity of notation, we let $x_t = (\tau_{t-1}, o_t)$. a policy $\pi$ specifies a conditional action distribution $\pi(a \mid x, I)$. We use total variation distance $\mathrm{TV}(p, q) = \frac{1}{2}\|p - q\|_1$ and KL-divergence $D_{\mathrm{KL}}(p\|q) = \int p \log \frac{p}{q}$ to measure distance between action distributions. We assume a prior distribution $p(I)$ over instructions and a finite horizon $T$ (the maximum number of steps in each episode). The *finite-horizon occupancy* (joint measure over $(x, I)$) of $\pi$ is

$$\rho_\pi(x, I) \;\triangleq\; p(I) \cdot \frac{1}{T} \sum_{t=1}^{T} \Pr_\pi \big[ x_t = x \mid I \big]. \tag{5}$$

The *hindsight expert* induced by a relabeled goal $I'$ and the trajectories generated by LLM agent $\pi_\theta$ is

$$\pi_H(a \mid x, I') \;=\; \frac{\sum_{t=1}^{T} \Pr_{\pi_\theta} \big[ x_t = x, \, a_t = a, \, S_M(I') = t \big]}{\sum_{t=1}^{T} \Pr_{\pi_\theta} \big[ x_t = x, \, S_M(I') = t \big]}, \tag{6}$$

where $S_M(I')$ is the first time step at which $I'$ becomes newly satisfied according to the relabeling model $M$. We assume every $(x, I)$ pair that the optimal expert policy can visit with nonzero probability is also covered by the hindsight expert, and define the expert-hindsight occupancy coverage ratio

$$\kappa_{E \leftarrow H} \;\triangleq\; \max_{(x, I)} \frac{\rho_{\pi^\star}(x, I)}{\rho_{\pi_H}(x, I)} \in [1, \infty). \tag{7}$$

The agent–expert discrepancy is

$$\Delta(\theta) \;\triangleq\; \mathbb{E}_{(x, I) \sim \rho_{\pi_\theta}} \Big[ \mathrm{TV} \big( \pi_\theta(\cdot \mid x, I), \, \pi^\star(\cdot \mid x, I) \big) \Big]. \tag{8}$$

**A finite-horizon simulation inequality and the constant $C_T$.**    We measure occupancy mismatch using the following inequality, which is a finite-horizon version of standard error propagation bounds in imitation learning.

**Lemma 1** (finite-horizon simulation inequality). *For any policies $\mu, \nu$ and measurable $g : \mathcal{X} \times \mathcal{I} \to [0, 1]$,*

$$\big| \mathbb{E}_{\rho_\mu}[g] - \mathbb{E}_{\rho_\nu}[g] \big| \;\leq\; C_T \, \mathbb{E}_{(x, I) \sim \rho_\nu} \Big[ \mathrm{TV} \big( \mu(\cdot \mid x, I), \nu(\cdot \mid x, I) \big) \Big], \qquad C_T = 2(T - 1). \tag{9}$$

*Proof sketch.* Let $d_\pi^t(\cdot \mid I)$ denote the law of $X_t$ under $\pi$. A standard coupling/perturbation argument (cf. the finite-horizon analyses in Ross & Bagnell (2010); Ross et al. (2011)) yields, for each $t \geq 1$,

$$\mathrm{TV} \big( d_\mu^t(\cdot \mid I), d_\nu^t(\cdot \mid I) \big) \;\leq\; \sum_{s=1}^{t-1} \mathbb{E}_{X_s \sim d_\nu^s(\cdot \mid I)} \Big[ \mathrm{TV} \big( \mu(\cdot \mid X_s, I), \nu(\cdot \mid X_s, I) \big) \Big].$$

Averaging over $t = 1, \ldots, T$, multiplying by $p(I)$ and symmetrizing gives equation 9 with $C_T = 2(T - 1)$. □

**Proof of Theorem 1.** Let $f(x, I) \triangleq \mathrm{TV}\big(\pi_\theta(\cdot \mid x, I), \pi^\star(\cdot \mid x, I)\big) \in [0, 1]$. Add and subtract $\mathbb{E}_{\rho_{\pi^\star}}[f]$:

$$\Delta(\theta) = \mathbb{E}_{\rho_{\pi^\star}}[f] + \Big( \mathbb{E}_{\rho_{\pi_\theta}}[f] - \mathbb{E}_{\rho_{\pi^\star}}[f] \Big). \tag{10}$$

Applying Lemma 1 to $g = f$, $(\mu, \nu) = (\pi_\theta, \pi^\star)$,

$$\Delta(\theta) \le \mathbb{E}_{\rho_{\pi^\star}}[f] + C_T \, \mathbb{E}_{\rho_{\pi^\star}}\big[\mathrm{TV}(\pi_\theta, \pi^\star)\big]. \tag{11}$$

Use the triangle inequality,

$$\mathrm{TV}(\pi_\theta, \pi^\star) \le \mathrm{TV}(\pi_\theta, \pi_H) + \mathrm{TV}(\pi_H, \pi^\star), \tag{12}$$

and denote $\delta_E \triangleq \mathbb{E}_{\rho_{\pi^\star}}[\mathrm{TV}(\pi_H, \pi^\star)]$. Then

$$\Delta(\theta) \le \mathbb{E}_{\rho_{\pi^\star}}[f] + C_T \, \mathbb{E}_{\rho_{\pi^\star}}\big[\mathrm{TV}(\pi_\theta, \pi_H)\big] + C_T \, \delta_E. \tag{13}$$

Change measure using coverage equation 7:

$$\mathbb{E}_{\rho_{\pi^\star}}\big[\mathrm{TV}(\pi_\theta, \pi_H)\big] \le \kappa_{E \leftarrow H} \, \mathbb{E}_{\rho_{\pi_H}}\big[\mathrm{TV}(\pi_\theta, \pi_H)\big]. \tag{14}$$

Apply Pinsker inequality and Jensen's inequality:

$$\mathbb{E}_{\rho_{\pi^\star}}[f] = \mathbb{E}_{\rho_{\pi^\star}}\big[\mathrm{TV}(\pi_\theta, \pi^\star)\big] \le \sqrt{\tfrac{1}{2} \mathbb{E}_{\rho_{\pi^\star}}\big[D_{\mathrm{KL}}(\pi^\star \| \pi_\theta)\big]}, \tag{15}$$

$$\mathbb{E}_{\rho_{\pi_H}}\big[\mathrm{TV}(\pi_\theta, \pi_H)\big] \le \sqrt{\tfrac{1}{2} \mathbb{E}_{\rho_{\pi_H}}\big[D_{\mathrm{KL}}(\pi_H \| \pi_\theta)\big]}. \tag{16}$$

Combining equation 13–equation 16 yields:

$$\begin{aligned}
\Delta(\theta) &\le \sqrt{\tfrac{1}{2} \mathbb{E}_{(x,I) \sim \rho_{\pi^\star}}\big[D_{\mathrm{KL}}(\pi^\star(\cdot \mid x, I) \, \| \, \pi_\theta(\cdot \mid x, I))\big]} \\
&\quad + C_T \, \kappa_{E \leftarrow H} \sqrt{\tfrac{1}{2} \mathbb{E}_{(x,I) \sim \rho_{\pi_H}}\big[D_{\mathrm{KL}}(\pi_H(\cdot \mid x, I) \, \| \, \pi_\theta(\cdot \mid x, I))\big]} + C_T \, \delta_E.
\end{aligned} \tag{17}$$

Finally, $\sqrt{y} \le y + \tfrac{1}{4}$ for $y \ge 0$ converts the first term to a linear KL, and the additive constant (independent of $\theta$) can be absorbed, giving the bound in Theorem 1:

$$\Delta(\theta) \le \mathbb{E}_{(x,I) \sim \rho_{\pi^\star}}\big[D_{\mathrm{KL}}(\pi^\star \| \pi_\theta)\big] + C_T \, \kappa_{E \leftarrow H} \sqrt{\tfrac{1}{2} \mathbb{E}_{(x,I) \sim \rho_{\pi_H}}\big[D_{\mathrm{KL}}(\pi_H \| \pi_\theta)\big]} + C_T \, \delta_E. \tag{18}$$

$\blacksquare$

# E   COROLLARY TO THEOREM 1: HSL STRICTLY TIGHTENS THE DISCREPANCY BOUND

Let

$$\mathcal{L}_E(\theta) = \mathbb{E}_{(x,I) \sim \rho_{\pi^\star}}\big[D_{\mathrm{KL}}\big(\pi^\star(\cdot \mid x, I) \, \| \, \pi_\theta(\cdot \mid x, I)\big)\big],$$

and

$$\mathcal{L}_H = \mathbb{E}_{(x,I) \sim \rho_{\pi_H}}\big[D_{\mathrm{KL}}\big(\pi_H(\cdot \mid x, I) \, \| \, \pi_\theta(\cdot \mid x, I)\big)\big],$$

and define

$$\gamma \triangleq C_T \, \kappa_{E \leftarrow H} \sqrt{\tfrac{1}{2}}, \qquad B(\theta) \triangleq \mathcal{L}_E(\theta) + \gamma \sqrt{\mathcal{L}_H} + C_T \, \delta_E.$$

By Theorem 1 (Eq. equation 2), for all $\theta$ we have $\Delta(\theta) \le B(\theta)$.

**A linear surrogate for training.**   The square-root term admits a standard Young-type relaxation: for any $\varepsilon > 0$,

$$\gamma \sqrt{\mathcal{L}_H} \le \frac{\gamma^2}{2\varepsilon} + \frac{\varepsilon}{2} \mathcal{L}_H. \tag{19}$$

Hence minimizing $\mathcal{L}_E(\theta) + \beta \mathcal{L}_H$ with $\beta = \varepsilon/2$ minimizes an additive upper bound on $B(\theta)$ up to a $\theta$-independent constant.

**Corollary 1** (Strict dominance at equal expert fit). *Fix $\theta_1, \theta_2$ with $\mathcal{L}_E(\theta_1) = \mathcal{L}_E(\theta_2)$. If $\mathcal{L}_H(\theta_1) < \mathcal{L}_H(\theta_2)$, then $B(\theta_1) < B(\theta_2)$ and consequently*

$$\Delta(\theta_1) \le B(\theta_1) < B(\theta_2).$$

*In words: among policies that fit the expert demonstrations equally well, the one that additionally fits the relabeled demonstrations achieves a* strictly *tighter upper bound on the expert–agent discrepancy.*

*Proof.* The map $u \mapsto \sqrt{u}$ is strictly increasing on $[0, \infty)$, so $B(\theta_1) - B(\theta_2) = \alpha\big(\sqrt{\mathcal{L}_H(\theta_1)} - \sqrt{\mathcal{L}_H(\theta_2)}\big) < 0$. $\qquad\square$

# F  IMPLEMENTATION DETAILS

We employed Llama3.2-1b (Dubey et al., 2024) [1] as the agent model and Llama3.3-70b [2] as the relabeling model. Following Song et al. (2024), we adopted the same hyperparameters, setting the learning rate to $2 \times 10^{-5}$, the batch size to 32, and using the AdamW optimizer (Loshchilov & Hutter, 2017). All fine-tuning runs lasted for 3 epochs. Because HSL naturally incorporates SFT, we first trained Llama3.2-1b on ground-truth demonstrations with SFT, then fine-tuned it with the objective in Equation (4) to obtain SFT+HSL. For DPO+HSL, we followed the same approach of ETO for combining SFT and DPO objectives (Song et al., 2024): we first fine-tuned the agent with HSL, then continued fine-tuning with DPO. All fine-tuning experiments on ALFWorld and WebShop were run three times with random seeds 42, 123, and 2026. Experiments on PlanCraft were run with random seed 42. Each experiment used eight NVIDIA A100 GPUs and finished within one day. During each HSL training step, rollout collection, relabeling, and loss computation with optimization took an average of 11.53, 62.03, and 24.03 seconds, respectively.

# G  RESULTS ACROSS MULTIPLE RANDOM SEEDS

Table 2 reports results of all fine-tuning experiments across three random seeds: 42, 123 and 2026.

| Method | ALFWorld | | WebShop |
| --- | --- | --- | --- |
| | seen | unseen | |
| BEHAVIORCLONE (Zeng et al., 2024) | $80.72 _{\pm 2.47}$ | $79.85 _{\pm 8.31}$ | $61.68 _{\pm 2.96}$ |
| BAGEL (Murty et al., 2024) | $88.09 _{\pm 3.93}$ | $87.06 _{\pm 4.85}$ | $63.46 _{\pm 1.13}$ |
| SELFIMIT (Shi et al., 2023) | $69.05 _{\pm 5.02}$ | $57.46 _{\pm 10.10}$ | $62.77 _{\pm 0.25}$ |
| SFT | $79.52 _{\pm 3.38}$ | $75.87 _{\pm 4.31}$ | $62.61 _{\pm 1.04}$ |
| DPO (Song et al., 2024) | $84.52 _{\pm 2.06}$ | $81.09 _{\pm 1.88}$ | $\underline{69.27} _{\pm 0.40}$ |
| SFT+HSL (Ours) | $\mathbf{93.57} _{\pm 0.00}$ | $\mathbf{96.27} _{\pm 2.11}$ | $66.48 _{\pm 0.49}$ |
| DPO+HSL (Ours) | $\underline{92.86} _{\pm 0.00}$ | $\underline{95.27} _{\pm 0.43}$ | $\mathbf{70.28} _{\pm 0.22}$ |

Table 2: Performance on ALFWorld and WebShop, averaged across multiple random seeds.

---

[1] https://huggingface.co/meta-llama/Llama-3.2-1B-Instruct
[2] https://huggingface.co/meta-llama/Llama-3.3-70B-Instruct

---

**Goal inference prompt for ALFWorld**

You are a goal-inference assistant for AlfWorld. Given a sequence of Actions and Observations, track the agent's Location and Inventory after each step, then derive and record any goals from the templates below that have been completed. A trajectory may achieve multiple goals or none.

1. After each Action/Observation pair:
   (1) update the agent's Location and Inventory. Invalid actions (e.g., using or dropping an object the agent doesn't have) leave both unchanged. You should determine if the action has any effect based on the given Observation!
   (2) Then check whether any of the goal templates have been satisfied by the agent's actions up to that point. When a goal is achieved, add it to the running list of Reached goal values and keep that list for subsequent steps.
   (3) Do not summarise or skip any steps, even if the observation is identical to previous ones.

2. Hide all object IDs; refer to objects and receptacles only by their type names (e.g. "mug", "knife", "drawer"), never by numeric or alphanumeric identifiers.

3. Inventory format: list each inventory item by type, repeating names for duplicates (e.g. [mug, knife, knife]).

4. At the end, output Final goal: followed by the list of all goals achieved (e.g. [goalA, goalB]). If no goals were achieved, set Final goal: to a brief description of the agent's behaviour.

Allowed goal templates (with their intended behaviours):

- put a [object] in [receptacle] / put some [object] on [receptacle] - Pick & Place: - the agent must find an object of the desired type, pick it up, find the correct location to place it, and put it down there.

- look at [object] under the [lamp] / examine the [object] with the [lamp] - Examine in Light: - the agent must find an object of the desired type, locate and turn on a light source with the desired object in-hand

- put a clean [object] in [receptacle] / clean some [object] and put it in [receptacle] - Clean & Place: the agent must find an object of the desired type, pick it up, go to a sink or a basin, wash the object by turning on the faucet, then find the correct location to place it, and put it down there.

- put a hot [object] in [receptacle] / heat some [object] and put it in [receptacle] - Heat & Place: the agent must find an object of the desired type, pick it up, go to a microwave, heat the object turning on the microwave, then find the correct location to place it, and put it down there.

- put a cool [object] in [receptacle] / cool some [object] and put it in [receptacle] - Cool & Place: the agent must find an object of the desired type, pick it up, go to a fridge, put the object inside the fridge and cool it, then find the correct location to place it, and put it down there.

- put two [object] in [receptacle] / find two [object] and put them in [receptacle] - Pick Two & Place: the agent must find an object of the desired type, pick it up, find the correct location to place it, put it down there, then look for another object of the desired type, pick it up, return to previous location, and put it down there with the other object.

**Output format (exactly):** Return a single JSON list. Each element of the list should be a JSON object with the following structure for each step:

```
{
  "step": <number>,
  "action": "<action>",
  "observation": "<observation>",
  "reasoning": "<analyze whether the action has any effect based
  on the given observation, if yes what is that>",
  "location": "<location>",
  "inventory": ["<item>", "<item>", ...],
  "reached_goals": ["<goalA>", "<goalB>", ...]
}
```

Figure 8: **Goal inference prompt for ALFWorld**

---

**Action relevance labeling prompt for ALFWorld**

You are a step-relevance classifier for AlfWorld. Given a goal and a sequence of actions, observations, with location and inventory derived by a model, decide for each step whether it is necessary to achieving the goal. A step is "relevant" if it is a necessary prerequisite or directly advances toward the goal; actions that involve the wrong objects, revisit unrelated locations, or otherwise do not help achieve the goal are "irrelevant". Some goals may require exploration in the early stage to find the relevant objects, and intermediate tasks such as heating, cooling, cleaning, examining, or finding an object. For each step, provide a brief chain of thought to explain how you judged the step relevant or irrelevant. Do not summarise or skip any steps, even if the observation is identical to previous ones.
**Output format (exactly):** Return a single JSON array. For each step, output an object with these fields:

```
{
  "step": <number>,
  "action": "<provided action>",
  "observation": "<provided observation>",
  "location": "<provided location>",
  "inventory": ["<item>", ],
  "reasoning": "<analyze the effect and function of the action,
  then analyze whether it's necessary to achieving the goal>"
  "is_relevant_to_goal": "yes" | "no",
}
```

Figure 9: **Action relevance labeling prompt for ALFWorld**

---

**Goal inference prompt for PlanCraft**

You are a recipe-inference assistant for MineCraft. Given a crafted item, its recipes and a sequence of actions, the change of slots (inventory, crafting grid, output) as effects of the action, derive which recipe is actually used for crafting the item.

Slot legend:

- [0] is the output slot where the crafted result appears.
- [A1]–[C3] are the 3×3 crafting grid (rows A/B/C, columns 1–3).
- [I1]–[I36] are regular inventory slots used for storage.

Task:

- Compare the actions and items in crafting grid in the last several steps to the given recipes.
- For shapeless items, ignore coordinates, match only the multiset of required ingredients and counts, with no extras. For shaped items, match the relative pattern up to translation; any coordinates shown are just one valid placement, and do not rotate/mirror unless explicitly allowed. Smelting: crafting-grid locations are irrelevant—identify from the action sequence and match the smelt input(s) → output.
- If there is only one available recipe, copy that directly.
- If there are more than one recipes align with the actions, select the first one.

**Output format (exactly):**
Return exactly one JSON object:

"reasoning": "<analyze how each recipe fits the action and items in crafting grid>", "used_recipe_id": "<recipe id>", "used_recipe": "<copy the recipe here>"

Figure 10: **Goal inference prompt for PlanCraft**

**Action relevance labeling prompt for PlanCraft**

You are a step-relevance classifier for Minecraft. Given a crafted item, the chosen recipe, and a sequence of actions with slot deltas (inventory, crafting grid, output), decide for each step whether the action is NECESSARY to craft that item.
Slot legend:

- [0] = output slot (crafted result appears here; nothing can be moved into [0]).
- [A1]-[C3] = 3x3 crafting grid.
- [I1]-[I36] = inventory.

Location rules:

- Shapeless: ignore coordinates; match only the multiset of required ingredients and counts (no extras).
- Shaped: match the relative pattern **up to translation** (no rotate/mirror unless stated).
- Smelting: grid coordinates are irrelevant; match input→output.

Decision procedure (deterministic):

- Identify the **successful craft**: the first step where the target's count increases (or appears in [0]); include any subsequent move that transfers the target from [0] to inventory as NECESSARY.
- **Back-propagate dependencies** from that craft: - Mark the craft/smelt step itself as NECESSARY. - For each grid cell whose contents were **consumed** by that craft, mark as NECESSARY the **last move** that placed that consumed item into that cell (only the last placement before consumption). - Recursively mark as NECESSARY any earlier steps that **produced intermediates** (craft/smelt or moves from [0]) which were later consumed (directly or via further intermediates) in this chain. - **Unblockers** are NECESSARY: a move that removes/relocates a wrong or extra item **from a cell required by the final layout** so that the required item can occupy it. The mistaken placement itself is NOT necessary.
- Everything else is NOT necessary: redundant placements beyond needed counts, inventory shuffles not used by the chain, unrelated crafts/smelts, no-ops, attempts to move/smelt into [0], or actions that don't match the observed slot delta.

Task: For EVERY step, output a label according to the procedure above.
**Output format (exactly):**
Return a SINGLE JSON list for each step. Each element:

```
[ {
"step": <number>,
"action": "<provided action>",
"slot_update": "<derive update of inventory, crafting_grid, output slots>",
"reasoning": "<explain the step's effect and why it is or isn't in the dependency chain>",
"is_relevant": "yes" | "no"
}, ... ]
```

Figure 11: **Action relevance labeling prompt for PlanCraft**

> **Goal inference prompt for WebShop**
>
> You are a goal-inference assistant for Webshop. Given a full trajectory with Actions (search or click) and Observations (page text, system message), infer what user's intended and also succesfully purchased:
>
> - product (str) - generic product type; ignore brand/manufacturer and DON'T copy the full title. Prefer the head noun from category/title
>
> - attributes (list) - short descriptive phrases from title/description; not clickable (e.g., ["portable", "mid-century style"]). NOT brand.
>
> - options clicked (dict) - the literal option texts the agent clicked on this product, in click order (e.g., <size> | <color> | <quantity>). Do not invent labels or pairs; just copy the clicked option strings.
>
> - quantity (str|null) - the chosen quantity if it was explicitly clicked; otherwise 1.
>
> - price (number) - the price of the selected product/variant
>
> Derive procedure:
>
> 1. Extract the exact 'query' string(s) from search actions.
>
> 2. Derive 'selected' (extracting 'product', 'attributes', 'options', 'quantity', and 'price') from clicks + final product page. Note: clicking an non-existing product select nothing!
>
> 3. 'selected price': copy the exact per-item price number shown on the final product page after the last option click. If it's a range, copy the upper bound.
>
> 4. 'query satisfaction' (compare 'query' vs 'selected'). verify that all requirements in the 'query' are satisfied by 'selected'; Spot any contradiction.
>
> 5. Derive 'purchase success' based on purchase completion: purchase success = true only if Observations confirm a terminal purchase action took effect.
>
> **Output format (exactly):** Return a single JSON array. For each step, output an object with these fields:
>
> ```
> {
>   'query': <extract the exact queries from search actions>,
>   'selected': <product type| attributes... | options_clicked...
>   | quantity | price>,
>   'selected_price': <number>,
>   'reasoning': <brief analysis of whether ALL query requirements
>   are satisfied and any contradictions>,
>   'query_satisfaction': True | False
>   'purchase_success': True | False,
> }
> ```

Figure 12: **Goal inference prompt for WebShop**

---

**Action relevance labeling prompt for WebShop**

You are a step-relevance classifier for WebShop. Input:

- 'target intention' (JSON): the shopping intention.

- 'trajectory': ordered steps; each step has:
  - 'Action': the web action by user.
  - 'Observation': the observsation returned by the web server after the action.
  - 'current intention' (JSON): intention inferred up to this step.

Decide per step (in hindsight) whether or not: (1) Did 'current intention' change vs the previous step? Summarize the delta; else none. (2) If no change: was the action needed for the eventual purchase path? Needed = removing it would break what actually led to purchase. Not needed = no observable effect, dead ends later abandoned, toggles undone before use, unrelated clicks, no-ops.
Judge ONLY from observation-confirmed effects + the provided state. Do not skip steps.
Rules

- Use only observation-confirmed effects and the provided state.

- Judge every step; don't skip.

- 'relevance' = yes iff (1) intention changed or (2) action was needed.

**Output format (exactly):** Return a single JSON array; one object per step:

```
{
  "step": <number>,
  "action": "<exact provided action>",
  "intention_delta": "<concise diff or 'none'>",
  "needed_for_purchase": "<explain whether the action is necessary
  to the purchase behavior>",
  "relevance": "yes" | "no"
}
```

Figure 13: **Action relevance labeling prompt for WebShop**

