# OpenReview forum: "Spinning Straw into Gold: Relabeling LLM Agent Trajectories in Hindsight for Successful Demonstrations"
_ICLR.cc/2026/Conference — ICLR 2026 Poster_

### Official Review · Reviewer_8bBB · 2025-10-30

**Soundness:** 3
**Presentation:** 3
**Contribution:** 1
**Rating:** 4
**Confidence:** 2

**Summary:**

This paper proposes a sample-efficient learning method that reuses agent trajectories via hindsight experience relabeling. The methods are benchmarked in embodied navigation tasks with good empirical performances. Theoretical performance bounds are provided for the proposed method as well.

**Strengths:**

The paper is easy to follow, and the introduced algorithms are simple to add on to existing models, making them widely usable. The baselines are chosen well and extensively, while the studies on sample efficiency, ablations, and analysis are also designed well. Empirical performances are promising for the proposed method. (especially section on ensuring the performance gain doesn't only come from using a more powerful LLM)

**Weaknesses:**

- None of the main results have any statistical significance or uncertain qualification. The tasks from ALFWorld and WebShop should ideally be reran multiple times for every algorithm in order to at least get some standard deviation in score or success rate.
- The novelty of this work is rather limited, as hindsight experience relabeling is a widely used technique across robot learning and reinforcement learning.

**Questions:**

see weaknesses

---

> ### Author Response · Authors · 2025-11-30
> **Response to review**
>
> We sincerely thank the reviewer for the time and feedback. We address each concern below.
>
> 1. **Experiment should reran multiple times**
>
> > *… The tasks … should ideally be reran multiple times for every algorithm…*
>
> Thank you for raising this concern. In the original submission, each method was run once due to the high computational cost of training full LLM agents and evaluating a large set of baselines. In response to your comment, we have now run two additional independent seeds for every training method, and we report mean ± standard deviation for all training methods in Table 1.
>
>
>
>
> 2. **Novelty limited: hindsight experience relabeling is used across robotics and RL**
>
>
> > *… hindsight experience relabeling is a widely used technique across robot learning and reinforcement learning.*
>
> Thank you for pointing out the application of HER in RL and robotics. We fully agree that hindsight relabeling is a well-established idea in those domains, and we cite these works in L124–L133. However, this idea is under-explored in the context of LLM agents. Our contribution is not the generic notion of relabeling, but extending it to LLM post-training for agentic tasks, where both the problem setting and the required methodology differ substantially from prior HER-style work.
>
> (1) **Problem setting**: Most HER-style methods operate in fully observed state spaces with explicit goal representations, where goal achievement is checked by a simple, deterministic predicate over the state. By contrast, LLM agents act in partially observable environments with high-level observations (e.g., text, GUI layouts); in our experiments, these are text-only benchmarks (ALFWorld, WebShop). Inferring what goals were achieved requires reasoning over long sequences of free-form textual observations. As a result, “relabeling goals” is no longer a trivial state-matching operation: it is a semantic inference task.
>
> (2) **Methodology**: Because goals cannot be read off from states, we use an auxiliary LLM to infer all achieved natural-language goals along the trajectory and to classify which actions were relevant, which prior HER methods do not need. Further, prior HER work in RL trains on relabeled data using the same RL objective. In contrast, our method is designed for LLM post-training, where SFT is more lightweight and stable, and it can flexibly integrate with any post-training algorithm.
>
> (3) **Impact for future work**: Last but not least, we propose and validate a key design intuition: a strong LLM is more reliable as a hindsight relabeler than as the acting agent itself. More broadly, this supports a practical design pattern for future LLM agents: use strong proprietary models as offline hindsight relabelers and distill their judgments into smaller deployed agents. This is relevant both for research on data-efficient post-training and for real-world deployments where expert demonstrations are expensive to collect.

---

### Official Review · Reviewer_jNmR · 2025-10-30

**Soundness:** 1
**Presentation:** 3
**Contribution:** 3
**Rating:** 2
**Confidence:** 5

**Summary:**

The paper proposes Hindsight Supervised Learning (HSL) for (text-only) LLM agents in closed-world environments (i.e., with a structured schema of available goals): in addition to training the policy on expert demonstrations, train on on-policy rollouts labeled on the fly by an auxiliary, highly capable LLM, which relabels rollouts with all goals actually achieved, as judged by the auxiliary LLM. Relabeled trajectories are kept in a FIFO queue that is continually updated, and enter training alongside pre-existing expert demonstrations. Theoretical results provide some motivation for their approach; experiments on ALFWorld and WebShop show meaningful gains on the former and smaller gains on the latter. The paper does not establish results for open-world goal spaces.

**Strengths:**

Practical, simple idea that composes known ingredients (hindsight relabeling + SFT) into a new, tidy, reusable pipeline for language agents.

Clear empirical signal on ALFWorld; sample-efficiency curves indicate strong wins at lower demo budgets. Notably, on ALFWorld Unseen the method reaches ≈92.5% with 800 demos vs ≈82.8% for DPO with >3,200, indicating real demo-efficiency.

Implementation details (masking + reweighting + on-policy refresh) are intuitive and ablations suggest each adds value.

Theory is tidy (imitation-style discrepancy bound) and aligns with the empirical recipe—even if mostly motivational.

**Weaknesses:**

The paper has potential to be a nice contribution but has serious methodological shortcomings in its current presentation:

- No variance / seeds (critical). Table results and curves lack seeds, error bars, or confidence intervals. Delta could be within variance. A single cherry-picked seed is an absolute non-starter to get the paper accepted. (Addressing this point alone would change my score from 2 to 4; see questions below).

- Code / reproducibility (critical). Prompts are provided, but there’s no released code or relabeled dataset for third-party checking (and no determinism details). Uploading assets as a ZIP to OpenReview (or to some CDN that is anonymized) for inspection is a totally reasonable academic practice (especially if there are manually labeled results on the paper) that the authors should be expected to satisfy.

- Cost/efficiency underreported (important). The paper states fine-tuning wall-clock but does not clarify whether this includes the compute cost for continual relabeling with a large model, nor report the latter separately. Without this, “data efficiency” is unclear relative to real cost.

- Label-quality evidence is thin (important). The relabeler is validated with a very small human spot-check of only 50 trajectories.

- Heavy dependency on a powerful annotator (important). Relabeling uses a ~70B model for a 1B agent. There’s no sensitivity to smaller relabelers; conclusions may depend on annotator strength.

- The framing of "almost unsupervised" could be misread (important). The method requires ground-truth demonstrations and mixes them in every training step. It does not replace demos; the current evidence shows materially improved use of demos, certainly not demo-free learning. A modification of language here could be in order. The paper also does not compare clearly the compute cost of their method vs that of their baseline (i.e. using expert demonstrations only); not discussing trade-offs between / scaling laws of compute and dataset size limits the reach of their conclusions.

- Ablation confounds in RELABELFAILURE (preferable). Because that variant relabels only failed trajectories, it gets less data as the base improves. This confounds “algorithm” with “data volume.”

- Theory–practice gap (preferable). The bound depends on coverage/optimality constants that are not measured nor bounded experimentally; masking/reweighting are argued heuristically to help them, but this link isn’t empirically probed.

These evidence gaps drive my Soundness=1 and overall score. My take is that this paper is a valuable practical contribution with clear empirical signal on the right kind of tasks, but acceptance should be contingent on addressing the issues above; most notably, statistical significance. These are fixable and would substantially strengthen the paper.

**Questions:**

I found this paper interesting. Below are some questions/requests that I think that, if addressed, would substantially improve the quality of the paper.

- Seeds/variance: How many seeds per result? Seems only one. Please report mean±std (≥5) for Table 1 and all curves.

- Transparency: Will you release code, relabeling prompts, and the relabeled data?

- Relabeler sensitivity & cost: How does performance change with 3B/8B/70B relabelers? Report relabeling tokens, wall-clock, and total GPU hours per setting.

- Label validation at scale: can you label a substantially larger set of trajectories (e.g. 200) and/or add automatic predicate checks (where supported) to strengthen the statistical significance, and report confusion matrices per goal type?

- Ablation deconfounding: Please re-run a variant of RELABELFAILURE that (i) relabels all trajectories (successes included) and (ii) fixes the relabeled-sample budget across variants to disentangle algorithmic effect from data volume.

- Theory diagnostics: Any proxy measurements over training for coverage (e.g., diversity/goal-type coverage of hindsight set) and hindsight-expert quality (environment predicate agreement) to support the claimed mechanisms?

thank you!

---

> ### Author Response · Authors · 2025-11-30
> **Response to review (1/2)**
>
> We sincerely thank the reviewer for the time and feedback. We address each concern below.
>
> ### **Weakness 1 & Question 1: No variance / seeds**
>
> > *Table results and curves lack seeds… A single cherry-picked seed is an absolute non-starter…How many seeds per result…*
> Thank you for raising this concern.
>
> (1) In the original submission, each method was run once due to the high computational cost of training full LLM agents and evaluating a large set of baselines. Importantly, we did not cherry-pick any run: all methods used a fixed random seed (42) for fairness.
>
> (2) In response to your comment, we have now run two additional independent seeds for every training method, and we report mean ± standard deviation for all training methods in Table 1.
>
> ### **Weakness 2 & Question 2: Code/data release**
>
> > *…Will you release code, relabeling prompts, and the relabeled data?*
>
> Thanks for your question. The prompts are included in Appendix C. According to the company's policy, the code and data are currently under internal review for release approval. We will release them once approved.
>
> ### **Weakness 3: Cost/efficiency underreported**
>
> > *The paper states fine-tuning wall-clock but does not clarify whether this includes the compute cost for continual relabeling with a large model, nor report the latter separately. Without this, “data efficiency” is unclear relative to real cost…*
> Thank you for raising the concern.
>
> (1) The reported training time in L928 refers to “all fine-tuning experiments” (L927), which and hence includes runs with HSL.
>
> (2) To make the computational cost explicit, we additionally measure the time consumption of each component in the HSL pipeline. At each training step, rollout collection, relabeling, and loss computation with optimization take on average 11.53 s, 62.03 s, and 24.03 s, respectively. We include this information in L930.
>
> (3) We have also updated the Limitations section (L782-786) to explicitly note that HSL incurs an additional computational cost compared to SFT and DPO.
>
> (4) Our notion of “sample efficiency” throughout the paper refers specifically to reducing dependence on ground-truth expert demonstrations, which are usually the dominant cost in training LLM agents. We agree that HSL adds training compute; however, this does not diminish our contribution of improving sample efficiency. Trading extra training FLOPs for better performance is standard in LLM-agent training, where many methods also use large models during training (e.g., for reward signal or data synthesis) to reduce reliance on human supervision.
>
> ### **Weakness 4: Label-quality evidence is thin**
>
> > *The relabeler is validated … only 50 trajectories…  can you label a substantially larger set of trajectories (e.g. 200)...*
>
> Thank you for your suggestion.
>
> (1) The main claim we validate through experiments is that HSL can improve existing training pipelines such as SFT and DPO by continuously training on the relabeled data. The analysis in Section 4.4 is intended as supplementary evidence to help understand why HSL works, rather than as the primary evaluation.
>
> (2) Moreover, a strict and accurate automatic predicate-based assessment is not very flexible since there are multiple ways to express achieved goals. Therefore, we ran a small-batch yet affordable manual evaluation of the relabeled data.
>
> (3) Nonetheless, we have conducted an expanded manual evaluation following your suggestion. We randomly sample 200 relabeled demonstrations and check the correctness of the labels. 180 out of 200 were correct (90% accuracy), closely matching the originally reported 92% from the smaller check. We update Section 4.4 accordingly and plot the accuracy by each goal type (Fig. 6).

---

> ### Author Response · Authors · 2025-11-30
> **Response to review (2/2)**
>
> ### **Weakness 5 and Question 3: Reliance on powerful relabeler model**
>
> > *…Relabeling uses a ~70B model for a 1B agent… conclusions may depend on annotator strength…How does performance change with 3B/8B/70B relabelers?...*
>
> Thank you for the comment. As discussed in our limitations section, our method relies on the zero-shot reasoning capacity of the external LLM for relabeling, and hence we adopt a powerful 70b model. However, as also mentioned by Reviewer 8bBB, simply using a strong external model does not explain our gains. We directly compare against other baselines that use the same 70B model but in different roles (introduced and analyzed in L324-335 & L339-341), and these methods perform significantly worse.
>
> ### **Weakness 6: "Almost unsupervised" can be misread**
>
> > *The framing of "almost unsupervised" could be misread…*
>
> We would like to clarify that we didn’t claim HSL is "almost unsupervised". The phrase "almost unsupervised" in L209 specifically refers to the "Relabeling process". In L298-L301, we also remind the reader that HSL requires ground-truth demonstrations. To further avoid any misreading, we make it clearer in L209.
>
> ### **Weakness 7: RELABELFAILURE uses less data**
>
> > *Ablation confounds in RELABELFAILURE… that variant relabels only failed trajectories, it gets less data as the base improves. This confounds “algorithm” with “data volume.”...*
>
> (1) The purpose of including RELABELFAILURE is to mirror the standard HER-style baseline, which by design relabels only the final achieved goal of failed trajectories. The resulting reduction in relabeled data is therefore an inherent limitation of the HER paradigm itself, not an artifact of our implementation. In addition, RELABELFAILURE depends on explicit reward signals to determine which trajectories can be relabeled, whereas HSL does not.
> (2) To further decouple “algorithm” from “data volume,” we ran an explicit control experiment on WebShop where SFT+HSL used a substantially larger relabeling batch ($b=96$ instead of 18). The performance remained essentially unchanged (66.30 vs. 66.48), indicating that training with more relabeled demonstrations does not necessarily bring improvement.
>
> 8. **Lack of theory diagnostics**
>
> > *…masking/reweighting are argued heuristically to help them, but this link isn’t empirically probed…Any proxy measurements…?*
>
> Since we do not have access to the optimal policy, it is intractable to measure the expert--hindsight occupancy gap directly. Instead, the downstream performance of different HSL variants in our ablation studies can be regarded as a practical proxy for how close the learned agent is to the optimal policy. Under a perfect relabeler assumption, if the agent converges to the optimal policy $\pi^*$, then the induced hindsight expert also converges to $\pi^\*$, which in turn implies $\kappa_{E\leftarrow H}\rightarrow 1$ and $\delta_E \rightarrow 0$. Taken together, the superior performance of full HSL compared to ablations provides indirect yet theoretically consistent evidence that our learning techniques reduce $\kappa_{E\!\leftarrow H}$ and $\delta_E$.

---

### Official Review · Reviewer_MGUs · 2025-11-01

**Soundness:** 3
**Presentation:** 4
**Contribution:** 3
**Rating:** 8
**Confidence:** 3

**Summary:**

The authors introduce an approach Hindsight supervised learning which trains an LLM agent on relabeled data. A relabeler LLM is used to relabel the LLM agent's trajectories with the goals that the LLM actually achieved. The authors provide results showing that by doing so, the agent is able to outperform baseline approaches.

**Strengths:**

The paper is very easy to read and provides a thorough theoretical and experimental analysis. The results are strong and very promising.

**Weaknesses:**

The relabeler’s output space is manually constrained with environment-specific templates. It would be helpful to discuss how HSL would scale to domains without predefined goals or how to automatically infer such goal spaces.

The method assumes that any achieved goal is beneficial to learn from but in some situations unintended achievements may not align with user intent or task utility. It would be useful to discuss ways to filter or weight relabeled goals based on relevance to user-defined objectives.

It would be useful to discuss the distinction between successful but unintended goals and partial progress toward intended goals. The method seems to treat both similarly, but their learning value may differ.

**Questions:**

Could you please discuss the goal space and how this might be defined in environments that don't have clearly predefined goals

Could you discuss how incorrect labels might dealt with and how they impact downstream performance?

---

> ### Author Response · Authors · 2025-11-30
> **Response to review**
>
> We sincerely thank the reviewer for the time and feedback. We address each concern below.
>
>  ### **Weakness 1 & Question 1: How to handle tasks without predefined goals & automatically infer goal spaces**
>
> > *… how HSL would scale to domains without predefined goals or how to automatically infer such goal spaces*
>
> Thank you for your question.
>
> (1) If a large set of language instructions is available, we can employ an LLM to summarize the goal space conditioned on the instructions set before post-training begins.
>
> (2) Otherwise, we can weight the unintended goals according to its similarity with the original instruction, or use the external LLM to judge the trajectory according to the high-level task(e.g. GUI agent) and determine whether any meaningful goals (e.g., everyday tasks a human might perform) are achieved.
>
> (3) We update and include this discussion in the Limitations section (L803–L808) accordingly.
>
> ### **Weakness 2: Reweight achieved goal based on relevance to user-defined objectives**
>
> > *… The method assumes that any achieved goal is beneficial to learn from …*
>
> Thank you for pointing this out.
>
> (1) In most existing LLM agent benchmarks, including the tasks used in our work, each goal in the predefined goal space is equally valuable to learn. However, we agree that when such predefined goal spaces are unavailable, some achieved but unintended goals may have low utility or even be harmful.
>
> (2) To address these challenges, future work could evaluate the utility of unintended goals by comparing them with the user instructions and employing safeguard models to filter out or penalize malicious goals. We update and include this discussion in the Limitations section (L809-812) accordingly.
>
>  ### **Weakness 3: Distinction between unintended and intended goals**
>
> > *It would be useful to discuss the distinction between successful but unintended goals and partial progress toward intended goals…*
>
> Thank you for pointing this out. We agree that learning from the subgoal of the intended goal might accelerate the learning more compared to an unintended goal that is less valuable for the task. However, as mentioned above, both benchmarks in our work equally value each goal within the goal space, therefore we have not distinguished them. As stated in the Limitations section (L810-812), we leave scoring the achieved goals for future work.
>
> ### **Question 2: How incorrect labels impact agent performance & how to address**
>
> > *… how incorrect labels might dealt with…*
>
> Thank you for the question. According to our theoretical analysis, incorrect goal relabels make the hindsight expert drift from the true expert and put less probability mass where the expert actually operates, increasing the hindsight–expert mismatch $\delta_E$ and the coverage ratio $\kappa_{E\!\leftarrow H}$ in Eq. (2). $\delta_E$ adds a linear offset, while $\kappa_{E\!\leftarrow H}$ scales the second term linearly. Both effects grow with the horizon.
>
> As discussed in the Limitations section (L769), future work can explore techniques to fine-tune the external LLM for a more accurate and fine-grained goal relabeling.

---

### Official Review · Reviewer_PFk9 · 2025-11-01

**Soundness:** 3
**Presentation:** 2
**Contribution:** 3
**Rating:** 4
**Confidence:** 3

**Summary:**

This paper introduces Hindsight Supervised Learning (HSL), a post-training method for LLM agents. It leverages the unintended but successful goals achieved during an agent’s rollouts by relabeling trajectories with goals that were actually accomplished. These relabeled examples are then used for fine-tuning, with two key techniques: irrelevant-action masking and demonstration reweighting. Experiments on ALFWorld and WebShop benchmarks show HSL improves both supervised fine-tuning and DPO, achieving higher success rates with fewer expert demonstrations.

**Strengths:**

- The paper provides somewhat strong empirical validation, showing consistent performance gains and higher sample efficiency across multiple benchmarks.
- There is a theoretical property offered although it is difficult to parse due to presentation issues. I still appreciate the attempt to include a theoretical result, which is rare to see in papers with large foundation models.

**Weaknesses:**

Below are my comments in the order of appearance in the paper.

1- In addition to hindsight relabeling, the paper should also discuss learning from play data in robotics. In that line of research, it is common to define goals in play data and label the trajectories with those goals.
2- Around line 177-178, I was confused by why the paper uses both a goal state and a language instruction. I understand that it is desirable to have the function $\delta$ because it will be useful for detecting the agent reaching other goals, but then, maybe goal state is all the paper needs? Why also have an instruction? Having both seems redundant from an RL perspective.
3- In line 195, I believe the statement $K \leq T$ implicitly assumes the goals are mutually exclusive so the agent cannot achieve multiple goals at the same time. Perhaps this should be clarified earlier, because in reality an agent can achieve both "picking up a fruit" and "picking up a banana" at the same time, for example (or "closing the drawer" and "putting the mug in the drawer", etc.)
4- "improve agent optimality" is a weird phrase. If something is optimal, it cannot be improved by definition.
5- There is a system prompt that describes the space of valid goals. So the paper needs a definition for each goal. Perhaps something like "this set of states achieves this goal". But if it is a mapping, there would be no need for an LLM. So I assume, what is needed is a natural language description of each goal. This should be part of the problem statement.
6- In the theoretical analysis section, there is this variable $h$ in Equation 1, but it is not defined anywhere. This makes it very difficult to follow the theory. I assume it is history. But then, how can $\tau_{t-1}$ for different values of $t$ can be equal to $h$? The former has varying length depending on $t$ whereas the latter is fixed. Does this mean the probability term inside the summation is nonzero only for a single $t$? If so, writing it as a summation is unnecessarily complex. Perhaps this is not the case, and there is something I miss about $h$. Since it is not defined, I do not know what it can be.
7- The conclusion section should discuss when (for what tasks) this method can and cannot be applied.

**Questions:**

Please see my comments and clarification questions in the Weaknesses section above.

---

> ### Author Response · Authors · 2025-11-30
> **Response to review**
>
> We sincerely thank the reviewer for the time and feedback. We address each concern below.
>
> ### **Weakness 1: Related work in learning from play data in robotics**
>
> > *… should also discuss learning from play data in robotics…*
>
> Thank you for your suggestion. Due to the space limit, we mainly cite the most representative work of hindsight experience relabeling. As you suggested, we have added citations of learning from play in L132-135.
>
> ### **Weakness 2: Why the formulation uses both a goal state and a language instruction)**
>
> > *…why the paper uses both a goal state and a language instruction… Why also have an instruction?...*
>
> Thanks for your question.
>
> (1) The instruction is necessary since it is the only goal specification available to the agent. The agent never observes a goal state, which appears only for evaluation. We introduce both terms to make explicit the partial-observability nature of these tasks: the agent must interpret the instruction to infer what goal state it should try to reach, based solely on language and observations.
>
> (2) To further avoid confusion, we update L178 to clarify the connection between instruction and goal state, as well as their roles in the formulation.
>
>
> ### **Weakness 3: Assumption of mutually exclusive goals**
>
> > *…the statement  implicitly assumes the goals are mutually exclusive… this should be clarified earlier…*
>
> Thanks for pointing this out.
>
> (1) Yes, this assumption reflects a property of ALFWorld and WebShop: in these environments, only one new goal becomes satisfied at any given step. But this assumption is task-dependent, and our method does not rely on this assumption.
>
> (2) To avoid confusion, we removed this assumption from the method section.
>
> ### **Weakness 4: Phrasing of "improve agent optimality"**
>
> > *"improve agent optimality" is a weird phrase*
>
> Our intended meaning is that optimizing the agent towards the optimal policy. Accordingly, we revise the phrasing in L142 and L156 for clarification.
>
> ### **Weakness 5: Including goal space in problem statement**
>
> > *There is a system prompt that describes the space of valid goals. So the paper needs a definition for each goal…*
>
> Thank you for pointing this out. Now we include the goal description in the problem formulation (L179).
>
> ### **Weakness 6: Definition of $h$**
>
> > *…there is this variable \( h \) in Equation 1, but it is not defined anywhere… Does this mean the probability term inside the summation is nonzero only for a single \( t \)?...*
>
> Thank you for flagging this.
>
> (1) $h$ is not a variable but a particular realization of $\tau_{t-1}$, used so that the hindsight expert $\pi_H$ is defined without carrying an explicit time index $t$.
>
> (2) We agree that the summation can be dropped safely. We originally wrote the sum to make it explicit that $\pi_H$​ is the action distribution at a random stopping time $S_M​(I′)$ (the first time the relabeled goal fires). To address your concern, we simplify Eq. 1 for clarity.
>
> ### **Weakness 7: Conclusion with when this method can and cannot be applied**
>
> > *The conclusion section should discuss when (for what tasks) this method can and cannot be applied.*
>
> Thank you for the suggestion.
>
> (1) We agree that clarifying the applicability of HSL is important. Our paper discusses the scope and underlying assumptions of HSL across several sections: the introduction (L58–64) outlines when hindsight relabeling is beneficial; the empirical takeaways (L476) and the limitations section (L768–770) elaborate on the factors of task characteristics and settings in which HSL may be less effective.
>
> (2) Following your suggestion, we highlight the above points again in the conclusion section (L493-496).

---

### Meta-Review · Area_Chair_EUGG · 2026-01-09

**Summary:**

This paper introduces Hindsight Supervised Learning (HSL), a post-training method for LLM agents. It leverages the unintended but successful goals achieved during an agent’s rollouts by relabeling trajectories with goals that were actually accomplished. These relabeled examples are then used for fine-tuning. Experiments on ALFWorld and WebShop benchmarks show HSL improves both supervised fine-tuning and DPO, achieving higher success rates with fewer expert demonstrations.

Reviewer concerns were:
1. presentation/clarity concerns
2. Seeds and statistical correctness.
3. Code release and transparency.
4. Relabeler size/cost.
5. More evidence on dependence on label quality.
6. Novelty.

**Reviewer Concerns:**

The authors ran many more seeds and share a plan for code release. They clearly discuss the role of relabelers. In this case, I don't think novelty should be a primary concern, its in very different domains.

**Reviewer Scores:**

PFk9 should increase their score to a 6 with manuscript updates and clarifications.
MGUs would remain the same.
jNmR would increase to a 5 or 6 with the changes.
8bBB would increase to a 6 with the changes. I also don't think we should be rejecting paper on just saying "not novel enough"

Overall, the paper has a set of valuable insights and would help the community understand a different type of method for LLMs.

---

### Decision · Program_Chairs · 2026-01-26

Accept (Poster)